# Development of a reliable, sensitive, and convenient assay for the discovery of new eIF5A hypusination inhibitors

Oumayma Benaceur[1], Paula Ferreira Montenegro[2], Michel Kahi[1], Fabien Fontaine-Vive[2], Nathalie M. Mazure[1], Mohamed Mehiri[2], Frederic Bost[1], Pascal Peraldi[1]*

1 Université Côte d'Azur, Inserm U1065, C3M, Nice, France, 2 Université Côte d'Azur, CNRS UMR 7272, Institut de Chimie de Nice, Nice, France

☯ These authors contributed equally to this work.
* pascal.peraldi@univ-cotedazur.fr

## Abstract

eIF5A is a translation factor dysregulated in several pathologies such as cancer and diabetes. eIF5A activity depends upon its hypusination, a unique post-translational modification catalyzed by two enzymes: DHPS and DOHH. Only a few molecules able to inhibit hypusination have been described, and none are used for the treatment of patients. The scarcity of new inhibitors is probably due to the challenge of measuring DHPS and DOHH activities. Here, we describe the Hyp'Assay, a convenient cell-free assay to monitor eIF5A hypusination. Hypusination is performed in 96-well plates using recombinant human eIF5A, DHPS, and DOHH and is revealed by an antibody against hypusinated eIF5A. Pharmacological values obtained with the Hyp'Assay, such as the $EC_{50}$ of DHPS for spermidine or the $IC_{50}$ of GC7 for DHPS, were similar to published data, supporting the reliability of the Hyp'Assay. As a proof of concept, we synthesized four new GC7 analogs and showed, using the Hyp'Assay, that these derivatives inhibit hypusination. In summary, we present the Hyp'Assay; a reliable and sensitive assay for new hypusination inhibitors. This assay could be of interest to researchers wanting an easier way to study hypusination, and also a valuable tool for large-scale screening of chemical libraries for new hypusination inhibitors.

## 1. Introduction

Eukaryotic Initiation Factor 5A, eIF5A, is a peculiar protein [1, 2]. While its acronym indicates that it is an initiator of translation, it is also involved in elongation, termination of translation, and in the CAT (C-terminal Ala-/Thr-tRNA) tailing process in the ribosome quality control pathway [3]. It is conserved among species. In procaryotes, where it is named EF-P, it is involved in resolving ribosome stalling in difficult mRNA sequences such as poly-proline regions. Its function in eucaryotes is debated. Some authors proposed that, as observed in procaryotes, eIF5A is involved in the translation of a restricted number of mRNA, rich in polyproline [4]. Others have reported that eIF5A is a general translation factor, involved in the translation of most cellular mRNAs [5, 6].

**Data Availability Statement:** All relevant data are within the manuscript and its Supporting Information files.

**Funding:** This work was supported by the French Government INSERM, CNRS, La Ligue Nationale contre le Cancer, by AVIESAN, Institut Thématique Multi-Organismes Cancer appel à projet PCSI 2023 (#ASC23016ASA) to FB and through the UCAJEDI Investments in the Future project managed by the National Research Agency (ANR) with the reference number ANR-15-IDEX-01 to FB. We thank the Canceropôle Provence-Alpes-Côte d'Azur and the Provence-Alpes-Côte d'Azur Region for the financial support provided to the MetaboCell and MetaboPure projects. O.B., P.F.M and M.K are supported by a PhD grant from the "Ministère de l'enseignement supérieur et de la recherche". The funders had no role in study design, data collection and analysis, decision to publish, or preparation of the manuscript.

**Competing interests:** The authors have declared that no competing interests exist.

eIF5A is the only known protein that needs a post-translational modification, named hypusination, to be active. Hypusination involves two enzymes: deoxyhypusine synthase (DHPS) and deoxyhypusine hydroxylase (DOHH) (Fig 1A). First, DHPS transfers the butylamine moiety of spermidine to the $\varepsilon$-amino group of eIF5A Lys50. This reaction is dependent upon $NAD^+$ and produces deoxy-hypusinated eIF5A. The structure of the human eIF5A - DHPS complex has been resolved recently providing important information about the molecular basis of the interaction between the enzyme and its substrate [7]. Only one eIF5A molecule binds to a DHPS homo-tetramer. The interaction involved the hypusination loop of eIF5A and induces slight conformational changes in DHPS. Interestingly, although the structure of the archaeal homologs of eIF5A-DHS and its human counterpart share significant similarity, the stoichiometry is different since four "archaeal eIF5A" binds to one "archaeal DHPS" tetramer [8]. Then, DOHH, an iron-dependent enzyme, hydroxylates carbon 9 of deoxy-hypusinated eIF5A Lys50, producing hypusinated eIF5A. So far, the structure of eIF5A-DOHH complex has not been resolved. Hypusination allows eIF5A to bind to the empty E-site of ribosomes where it interacts with the peptidyl-tRNA at the P-site to facilitate translation [1].

EIF5A dysregulation is involved in several pathologies [2, 9]. (i) It is considered an oncogene since it is overexpressed in several cancers and is associated with a poor prognosis. Its inhibition slows the growth of cancer cells and reduces metastasis [10], and blockade of eIF5A hypusination limits colorectal cancer in mice [11]. (ii) Inhibition of eIF5A is considered as a potential therapeutic target in viral infection by HIV and Ebola virus. (iii) A decrease in eIF5A activity has been shown to improve glucose tolerance in diabetic mice. (iv) Reduction of eIF5A activity is beneficial for the treatment of ischemic injuries such as those occurring during organ transplantation and stroke. (v) More recently, inhibition of eIF5A hypusination has been proposed as a potential treatment for pulmonary arterial hypertension [12].

Consequently, the search for hypusination inhibitors is a major focus of research [13]. Some hypusination inhibitors, such as GC7, a well-described DHPS inhibitor, are available *in vitro* (as discussed later). However, none of them have reached the clinic. The limited number of known eIF5A inhibitors is probably correlated to the difficulty of measuring DHPS and DOHH activity, and the absence of an established assay for large-scale molecule screening.

The conventional DHPS assay is an *in vitro* experiment that requires recombinant, or purified, eIF5A and DHPS and radiolabeled [1,8-$^3$H] spermidine. [$^3$H]deoxyhypusinated-eIF5A is quantified using ion exchange chromatography or by a filter binding assay [14]. Although effective, this technique involves the use of radioactivity and HPLC equipment. Another method measures the amount of NADH produced during the deoxyhypusination reaction [15]. This technique is more convenient but is less sensitive and an indirect measurement of deoxyhypusination. Last, some authors have performed a quantitative NAD assay [16] followed by a qualitative hypusination assay by Western blot [7].

The measure of DOHH activity is even more challenging. [$^3$H]deoxyhypusinated-eIF5A is produced as described previously. After incubation with DOHH, proteins undergo acid hydrolysis and [$^3$H]deoxyhypusine and [$^3$H]hypusine are separated by ion exchange chromatography [17].

Here we described the Hyp'Assay, a new assay to measure both DHPS and DOHH activity. The principle of this assay is to carry out the hypusination of eIF5A in 96-well plates using recombinant eIF5A, DHPS, and DOHH. Hypusination of eIF5A is detected using a specific antibody, revealing the activity of DHPS and DOHH. Unlike existing assays, it can be implemented in most laboratories. It is reliable, fast, quantifiable, cost-effective, and as sensitive as assays that use radioactivity.

Using the Hyp'Assay we describe new inhibitors of hypusination. We show that, both in our cell-free assay and in intact cells, these compounds behave as hypusination inhibitors.

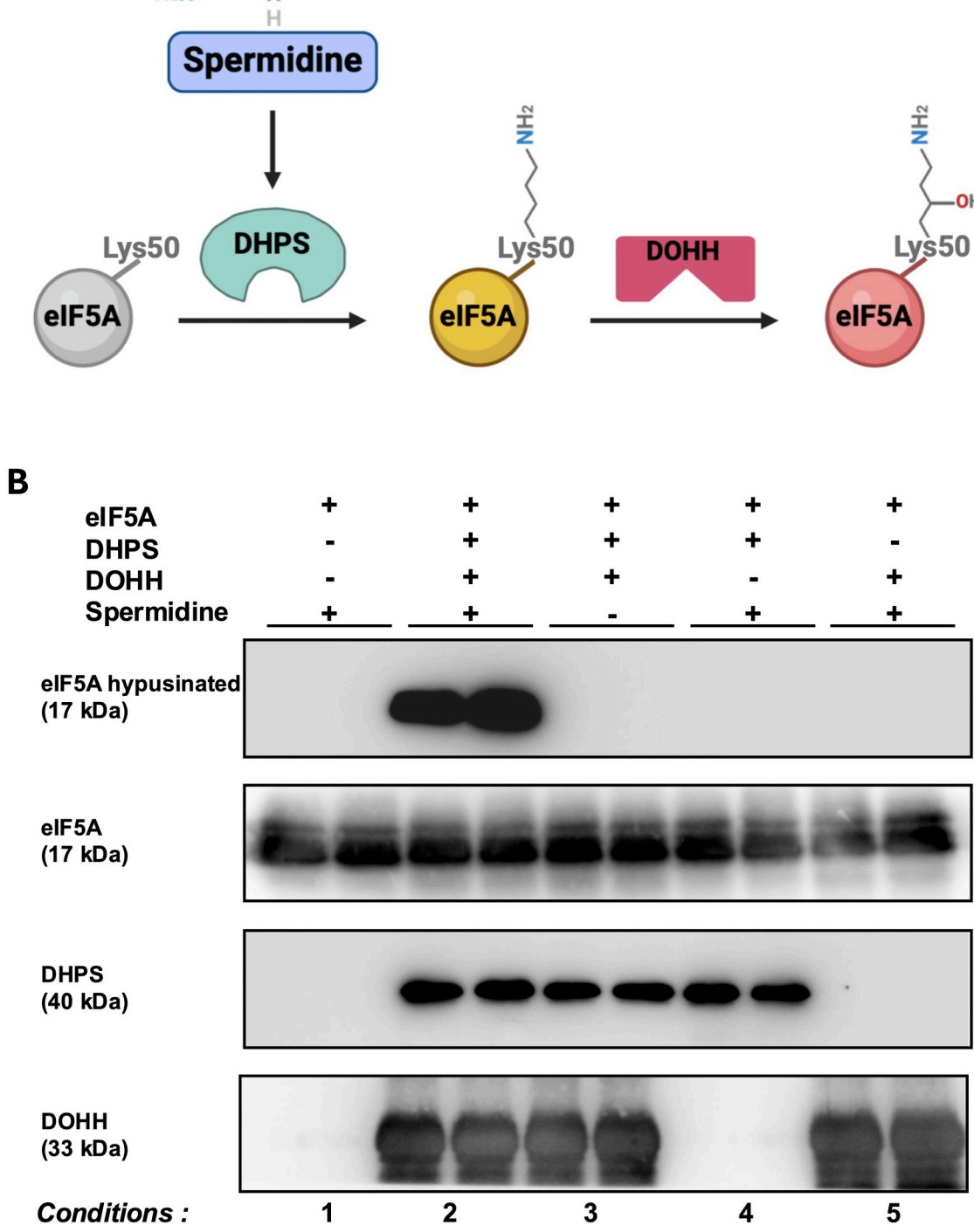

**Fig 1. Hypusination in a cell-free system.** (A) Reaction of eIF5A hypusination. (B) The hypusination reaction was performed in the presence of recombinants eIF5A (5 μg), DHPS (1 μg), DOHH (1 μg), and spermidine (100 μM) with or without one of the components. Proteins were analyzed by Western Blot with the indicated antibodies.

## 2. Results

### 2.1. Hypusination reaction in a cell-free system

We aimed to develop a cell-free system to measure the activity of the two enzymes necessary for eIF5A hypusination: DHPS and DOHH. To this end, we produced and purified the three partners of the reaction: eIF5A, DHPS, and DOHH as recombinant proteins (S1 Fig). Then, we performed an hypusination assay *in vitro*.

The hypusination of eIF5A is a two-step reaction (Fig 1A). First, eIF5A was incubated for 2 hours with DHPS, NAD$^+$, and spermidine in an appropriate buffer to induce eIF5A deoxyhypusination. Then, deoxyhypusinated eIF5A was hydroxylated by DOHH for 1 hour. Control incubations, without one of the components, were performed. These reactions were analyzed by Western blots (Fig 1B).

In the presence of all components (condition 2), eIF5A was heavily hypusinated. The absence of spermidine, DHPS, or DOHH led to a complete loss of hypusination. Unmodified eIF5A (conditions 1, 3, and 5) or deoxyhypusinated eIF5A (condition 4) was not detected in the anti hypusinated eIF5A Western blot, indicating that the antibody (Hpu24) is specific for hypusinated eIF5A as described in [18].

The sensitivity and specificity of the reaction indicated that the conditions were met to set up our hypusination assay in 96-wells.

### 2.2. Set-up of the Hyp'Assay

**2.2.1. The principle of the Hyp'Assay.**   The principle of the Hyp'Assay (Fig 2A) is to perform eIF5A hypusination reaction in 96-well plates, allow eIF5A to adsorb to the plate, incubate with an anti-hypusine antibody and a secondary antibody conjugated to HRP and reveal hypusination using TMB as a substrate.

**2.2.2. The Hyp'Assay is a quantitative and sensitive test.**   We performed a dose-response analysis of eIF5A to verify if the Hyp'Assay was quantitative. Increasing doses (0 to 6 μg in 50 μl) of eIF5A were hypusinated and revealed as described above. As observed in Fig 2B, a linear correlation was observed between the quantity of eIF5A and the hypusination signal. Then, we determined the optimal quantities of DHPS and DOHH. We performed a dose-response analysis using a fixed quantity of eIF5A (3 μg) and one of the enzymes DHPS or DOHH (2 μg) while different doses of the other enzyme (0 to 2 μg) were tested (Fig 2C). Both curves reached a plateau around 1 μg.

We studied the sensitivity of our assay to pH. Some authors performed hypusination reaction at pH 8.0 [19, 20] while others at pH 9.0 [21, 22]. This is supposedly because a higher pH increases the proportion of properly charged spermidine with the unprotonated secondary amino group. In the Hyp'Assay, hypusination was optimal at pH 8.0 and decreased around 50% at pH 7.5 or pH 9.0 (S2 Fig).

We performed the Hyp'Assay in the presence of an increasing concentration of spermidine (0 to 1000 μM) at pH 8.0 and pH 9.0 (Fig 2D). As above, hypusination was more efficient at pH 8.0 than at pH 9.0 for all tested concentrations of spermidine. The EC$_{50}$ of DHPS for spermidine at pH 8.0 was 5 μM, which is consistent with the previously published Km of human DHPS for spermidine; 4 μM [23] or 7.6 μM [24].

One of the purposes of the Hyp'Assay is to screen chemical libraries. Most of these libraries (Prestwick, Targetmol. . .) provide molecules in DMSO. We tested if this solvent could affect the Hyp'Assay by performing a dose-response analysis with an increasing percentage of DMSO (0 to 10% Fig 2E). At standard concentrations of DMSO (less than 10%), the hypusination of eIF5A was not affected. At the highest concentration of DMSO (10% of the total

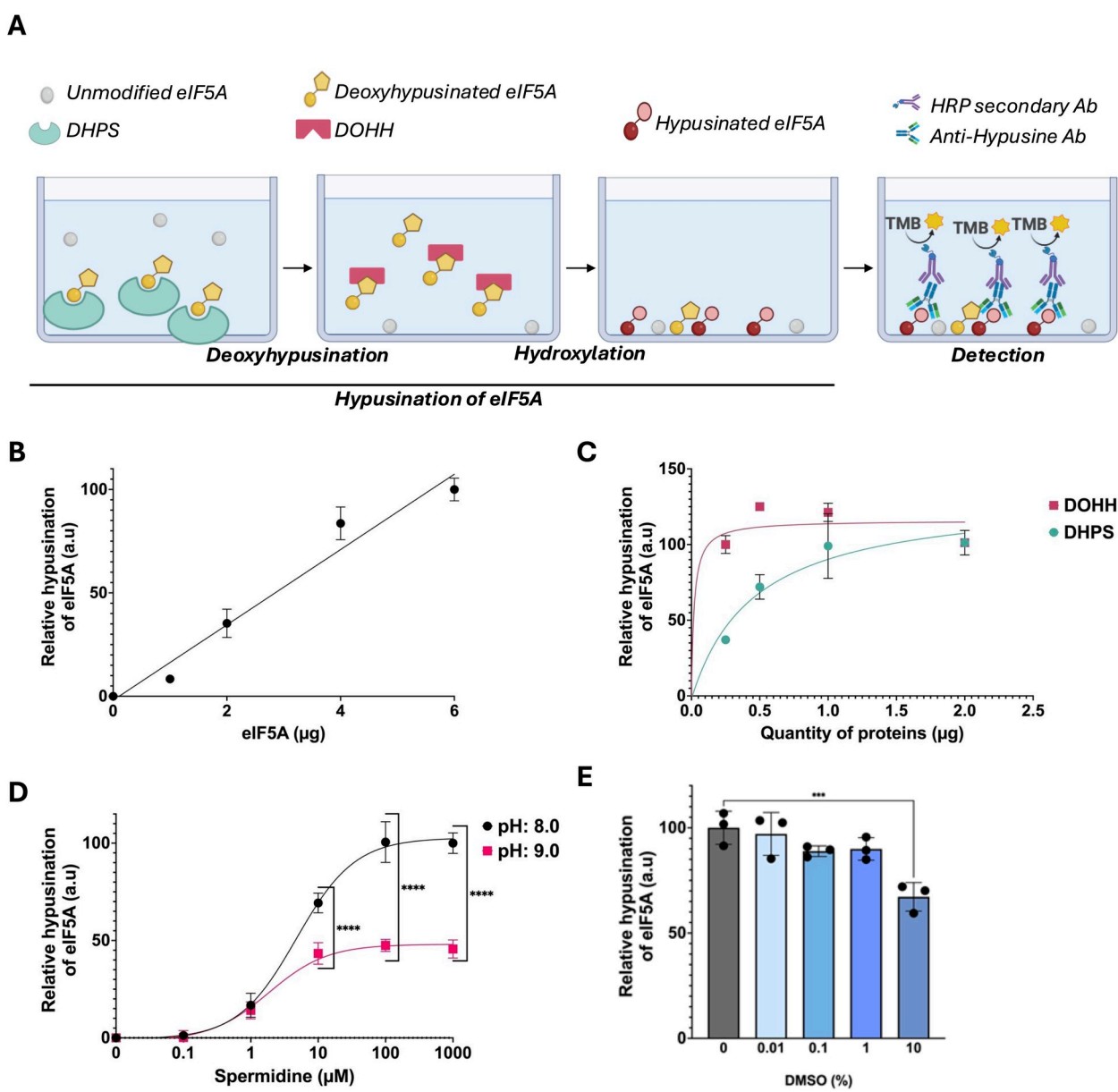

**Fig 2. Set-up of the Hyp'Assay.** (A) Principle of the Hyp'Assay. (B) Dose-response of the quantity of eIF5A to 0 from 6 μg with 2 μg of DHPS and DOHH and 100 μM of spermidine. (C) Dose-response of DHPS or DOHH with a fixed quantity of eIF5A (3 μg) and a fixed quantity of DOHH or DHPS (2 μg). (D) Dose-response of spermidine at two pH: 8.0 and 9.0 (E) Dose-response of DMSO in the reaction of hypusination.

reaction mixture), a 33% inhibition was observed. This indicates that DMSO, would not interfere with the screening of molecules with the Hyp'Assay.

Wątor et al [23], observed that spermine could be used as an alternate DHPS substrate. We performed a dose response experiment comparing spermine and spermidine in the Hyp'Assay. As observed in S3 Fig, spermine can be used as a substrate by DHPS in the Hyp'Assay, although with lower efficiency than spermidine. Under physiological conditions, with spermine concentration in the micromolar range and the difference of affinity between DHPS for spermine and spermidine, the contribution of spermine to eIF5A hypusination is probably low.

**2.2.3. Use of the Hyp'Assay to characterize hypusination inhibitors.** We tested the ability of known eIF5A hypusination inhibitors using the Hyp'Assay. First, we studied GC7 (N1-guanyl-1,7-diaminoheptane), the most used DHPS competitive inhibitor; a spermidine analog targeting the spermidine-binding site of DHPS [25]. A dose-response analysis of GC7 was performed (1 nM to 10 μM) at 100 μM and 5 μM of spermidine (Fig 3A). At 100 μM spermidine, the inhibitory effect of GC7 began at 0.1 μM and was maximal at 10 μM with an IC$_{50}$

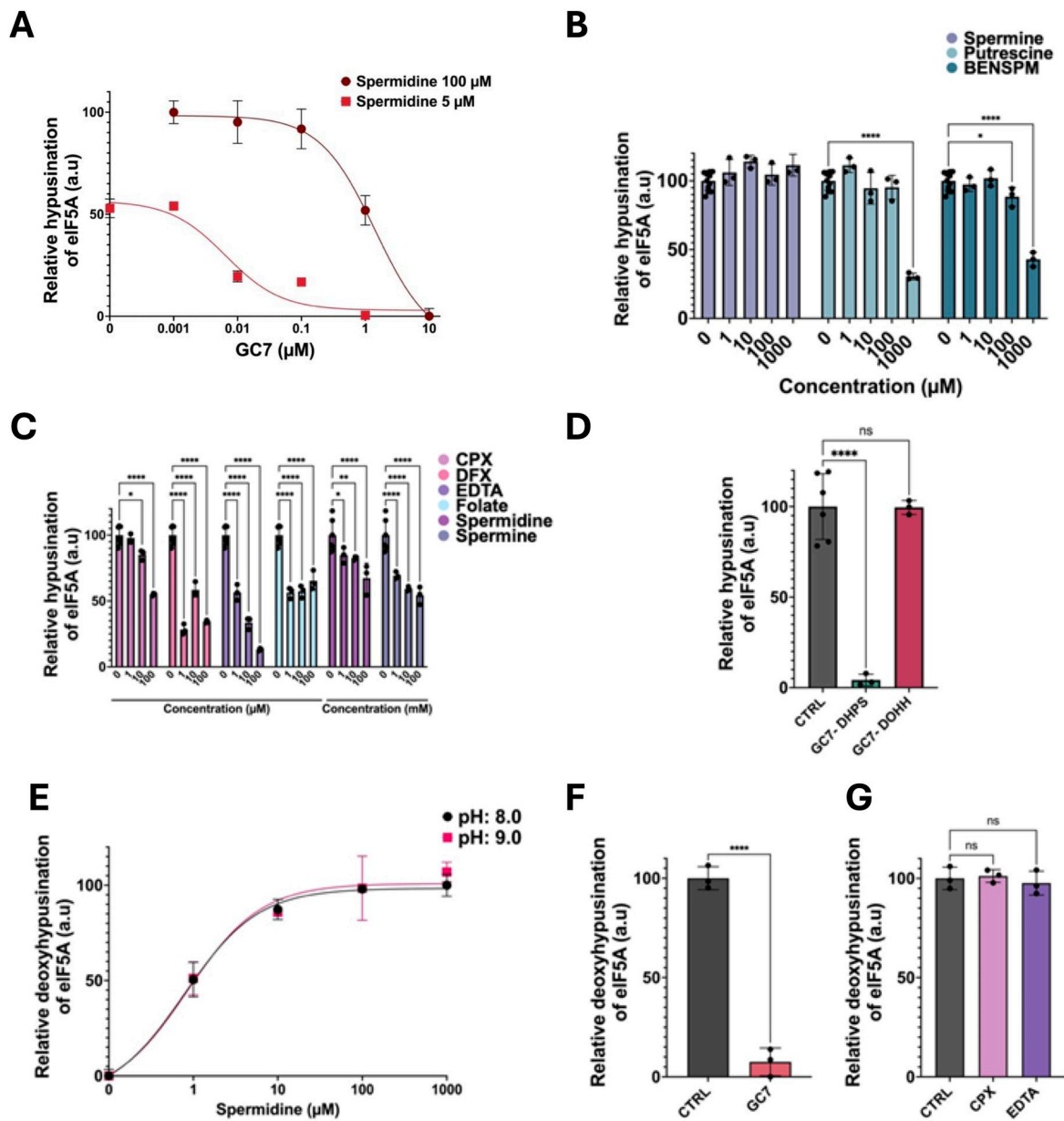

**Fig 3. Validation of the Hyp'Assay.** (A) Dose-response of GC7 to 0 from 10 μM with two concentrations of spermidine (5 and 100 μM). (B) Dose-response (0 to 1 mM) of different DHPS inhibitors; Spermine, Putrescine, or BENSPM. (C) Dose-response of different DOHH inhibitors (0 to 100 μM), CPX, DFX, EDTA, Folate, and Spermidine. (D) GC7 is added during the DHPS or the DOHH step in the reaction of hypusination with 5 μM of spermidine. (E-G) A reaction of deoxyhypusination of eIF5A is performed and revelated with Hpu98. (E) Dose-response of spermidine at two pH: 8.0 and 9.0. (F-G) GC7 (F), CPX or EDTA (G) are used in the reaction of deoxyhypusination of eIF5A with 5 μM of spermidine.

of 1.5 μM. At 5 μM spermidine, the inhibition was detectable at 0.01 μM of GC7 and was complete at 1 μM GC7 with an $IC_{50}$ of 6.8 nM. The difference in GC7 behavior at 5 and 100 μM spermidine is consistent with its mechanism of action as a competitive inhibitor. The $IC_{50}$ of GC7 determined with the Hyp'Assay (6.8 nm) at 5 μM spermidine falls within the range of the ones previously reported, 17 nM [25] and 50 nM [26].

Then, we realized dose-response analyses with three other polyamines; spermine, putrescine, and BENSPM (N1,N11-diethylnorspermine) an inducer of SSAT (Spermidine/spermine N 1 -acetyltransferase), a key enzyme in polyamine metabolism [27]. As observed in Fig 3B, concentrations of spermine, up to 1 mM, did not affect eIF5A hypusination while putrescine induced a 70% inhibition at 1 mM. These results are consistent with a previous report [25]. 100 μM of BENSPM induced a slight decrease in hypusination (12%) that reached 43% at 1 mM. To the best of our knowledge, this is the first observation of inhibition of hypusination by BENSPM.

We then tested the ability of DOHH inhibitors to inhibit eIF5A hypusination using the Hyp'Assay. During the Hyp'Assay, DOHH inhibitors were added after the deoxyhypusination reaction.

So far, no specific DOHH inhibitors have been characterized. DOHH is known to be inhibited by iron chelators such as ciclopirox (CPX), deferoxamine (DFX), and EDTA. Additionally, using computer simulation, Katiki et al. proposed that folic acid could inhibit DOHH [28]. We performed a dose-response analysis of these molecules with the Hyp'Assay (Fig 3C). The three irons chelators inhibited DOHH. CPX induced a maximal inhibition of 50%, which is consistent with what has been published for human recombinant DOHH [29]. EDTA and DFX were potent inhibitors with a maximal inhibition of 80% at 100 μM EDTA. Folate was also found to induce a 50% inhibition at 1 μM confirming, to some extent, the theoretical predictions.

Although these experiments show that the Hyp'Assay could be used to find new hypusination inhibitors, as it is, it cannot discriminate between DHPS and DOHH inhibitors. To address this issue, we compared the ability of GC7 to inhibit DHPS or DOHH by adding GC7 during the first step of the Hyp'Assay; when DHPS mediates the deoxyhypusination reaction (Fig 2A), or only during the second step, when DOHH hydroxylate deoxyhypusinated eIF5A. When GC7 was added during the first step it induced a total inhibition of hypusination (Fig 3D). By contrast, if GC7 was added only during the hydroxylation step no inhibition was observed. This confirms the specificity of GC7 towards DHPS. When finding new hypusination inhibitors adding them during the first or the second step of the Hyp'Assay will allow us to determine if the inhibition affects DHPS or DOHH. Another way to discriminate between DHPS or DOHH inhibitors would be to use Hpu98, an antibody that recognized both hypusinated and deoxyhypusinated eIF5A, instead of Hpu24, which is specific for hypusinated eIF5A [18]. This kind of approach has been utilized recently to demonstrate by Western blot that the putative hydroxylation site in DHPS from Trichomonas vaginalis is nonfunctional [30]. We performed a new Hyp'Assay by omitting DOHH in the reaction mix and using the Hpu98 antibody. First, we performed a dose response experiment with spermidine at pH: 8.0 and 9.0 (Fig 3E). The Hyp'Assay allows the detection of deoxyhypusinated eIF5A in a spermidine-dependent manner. Under these conditions, deoxyhypusination was equally efficient at both pH tested. This suggests that the decrease in activity at pH 9.0, observed in Fig 2D, is probably due to a decrease in activity of DOHH at pH 9.0. We also tested the ability of GC7 (Fig 3F) and CPX or EDTA (Fig 3G) to inhibit DHPS. As expected, only GC7, a DHPS inhibitor, was efficient in inhibiting deoxyhypusination. This provided us with another method to distinguish between DHPS and DOHH inhibitors.

Together, these results indicate that the Hyp'Assay can be used to study eIF5A hypusination inhibitors. We decided to challenge the Hyp'Assay to identify new inhibitors of hypusination.

**2.2.4. Description of GC7 analogs.** In the literature [26], several GC7 analogs, structurally related to the natural substrate spermidine, have been synthesized and evaluated for their capacity to inhibit DHPS. From a structural point of view, the most active derivatives feature: (i) an unbranched 7- or 8-carbon chain or a straight 7-member chain in which the central atom could be a heteroatom such as O, S, or N, (ii) a methyl substituent on the same carbon as the primary amino group at its aminopropyl end of spermidine, such as 1-methylspermidine, and (iii) basic groups, ammonium or guanidinium, at the extremities [26]. These observations have led to the construction of a model for the spermidine binding site within DHPS (Fig 4A). The primary amine groups of spermidine bind to zones **A** and **B**. The primary amine function

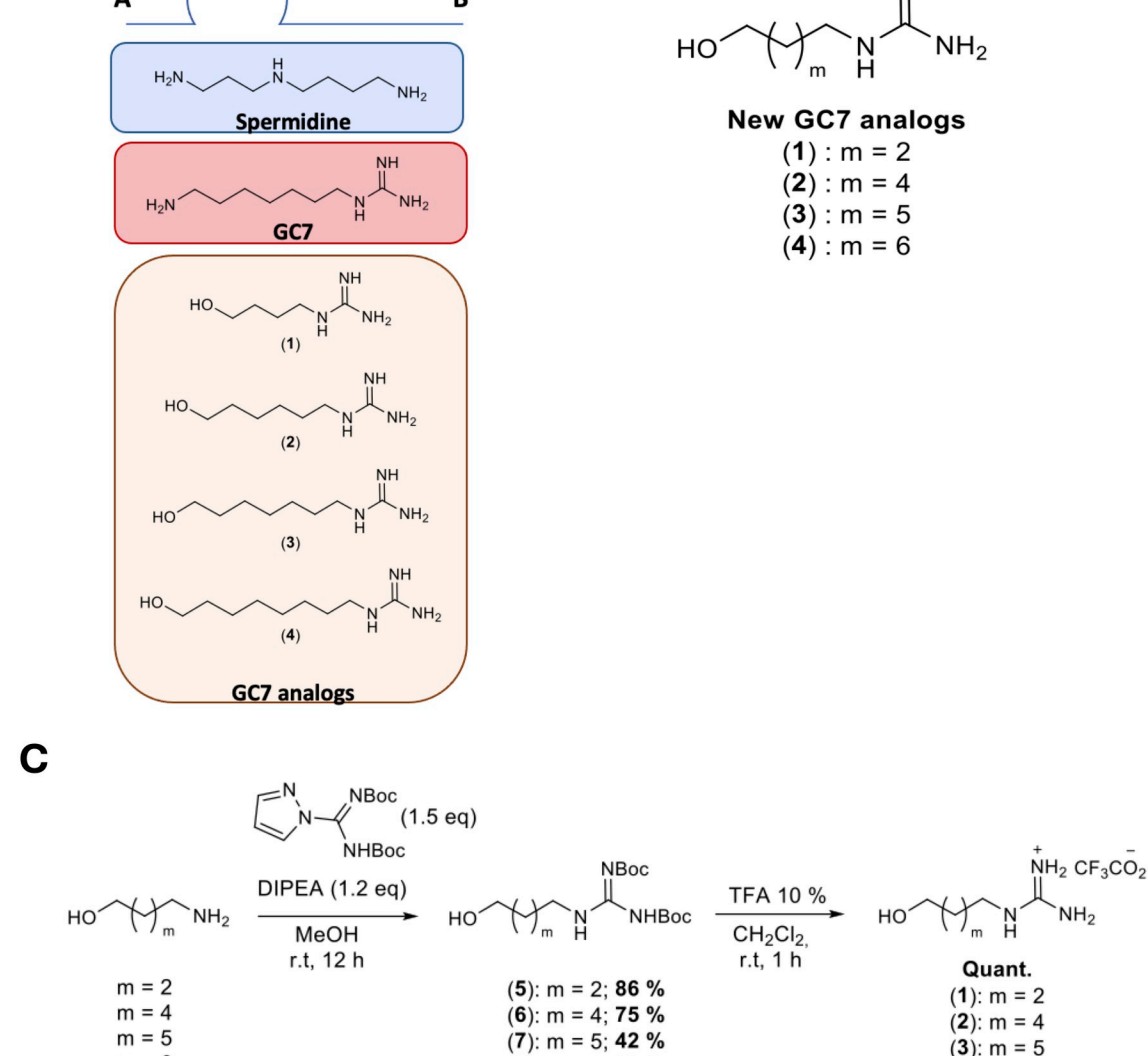

**Fig 4. Description of GC7 analogs.** (A) Model of spermidine and GC7 binding site within the DHPS active site. (B) GC7 analogs structures (structures depicted in neutral form). (C) GC7 analogs synthesis.

at the aminopropyl end of spermidine interacts with zone **A**, which preferentially accommodates an amino group. The aminobutyl group of spermidine is directed towards zone **B**, which preferentially accommodates a guanidinium group. Lastly, zone **C** represents the part of the binding site that specifically interacts with a methyl substituent located on the carbon bearing a primary amine function. Monoguanylated diamine derivatives exhibited better DHPS inhibition compared to diamines and diguanidines [26]. To date, no derivatives featuring both guanidinium and alcohol functionalities at the extremities have been reported for their DHPS inhibitory activities.

In this context, we synthesized in two steps the following derivatives, 4-hydroxybutylguanidine (**1**), 6-hydroxyhexylguanidine (**2**), 7-hydroxyheptylguanidine (**3**), and 8-hydroxyoctylguanidine (**4**) (Fig 4B) with an overall yield of up to 40%.

First, *N,N'*-di-*tert*-butoxycarbonyl-4-hydroxybutylguanidine (**5**), *N,N'*-di-*tert*-butoxycarbonyl-6-hydroxyhexylguanidine (**6**), *N,N'*-di-*tert*-butoxycarbonyl-7-hydroxyheptylguanidine (**7**), and *N,N'*-di-*tert*-butoxycarbonyl-8-hydroxyoctylguanidine (**8**) were obtained by reaction of 4-aminobutanol, 6-aminohexanol, 7-aminoheptanol (obtained by reduction of 7-aminoheptanoic acid with $LiAlH_4$), and 8-aminooctanol, respectively, with *N,N'*-diBoc-1*H*-pyrazole-1-carboxamidine in the presence of DIPEA in yields ranging from 42 to 86%. Finally, the Boc protecting groups of **5**–**8** were cleaved in acidic media with a solution of 10% TFA in dichloromethane to yield compounds **1**–**4** (Fig 4C).

**2.2.5. GC7 analogs are a new family of hypusination inhibitors.** We performed a dose-response of **1**, **2**, **3**, and **4** in the Hyp'Assay at 5 μM spermidine (Fig 5A). An inhibition of hypusination was detectable from 1 μM for **1**, **2**, and **3**, while the effect of **4** began at 100 μM. At 100 μM all GC7 analogs induced a 50% inhibition. Thus, the compounds are less efficient to inhibit hypusination than GC7, probably due to the -OH to -NH2 modification of the molecules (Fig 4A and 4B).

We tested the ability of the GC7 analogs to inhibit hypusination in intact cells. MCF-7 cells (Fig 5B), a classical model of breast cancer cell line, and HT29 cells (Fig 5C), a cell model of colorectal cancer, were treated with 40 μM of GC7, **1**, **2**, **3**, and **4** for 24h. Cells were lysed and analyzed by Western blot. As observed, none of the molecules modified eIF5A expression. In both cell lines, GC7 was the more potent molecule, inhibiting eIF5A hypusination by 80%. GC7 analogs inhibited hypusination from 35 to 55%. This indicates that GC7 analogs are a new class of DHPS inhibitors.

Together, these data indicate that the Hyp'Assay can be used to test and compare the ability of new molecules to inhibit hypusination.

## 3. Discussion

eIF5A is considered as a crucial player in several pathologies ranging from cancer to diabetes [2, 9, 10] making it an important therapeutic target for the development of inhibitors. eIF5A activity is controlled by its hypusination which depends upon two enzymes DHPS and DOHH. To date, no molecules that could directly inhibit the translational function of hypusinated eIF5A, for instance by interfering with the binding of eIF5A to the ribosome, have been described. Pharmacological studies have focused on DHPS inhibitors, and, to a lesser extent, DOHH. So far, few molecules have been discovered and none have reached the clinic.

The best-characterized DHPS inhibitor is GC7, a competitive inhibitor of spermidine discovered in 1993 [25]. Competitive DHPS inhibitors are not considered optimal candidates for clinical use [1, 10]. Indeed, it is estimated that the concentration of spermidine in cells is in the millimolar range while free spermidine is estimated to reach 200 μM [31]. Such an elevated concentration of intracellular spermidine could be a challenge for competitive inhibition. This

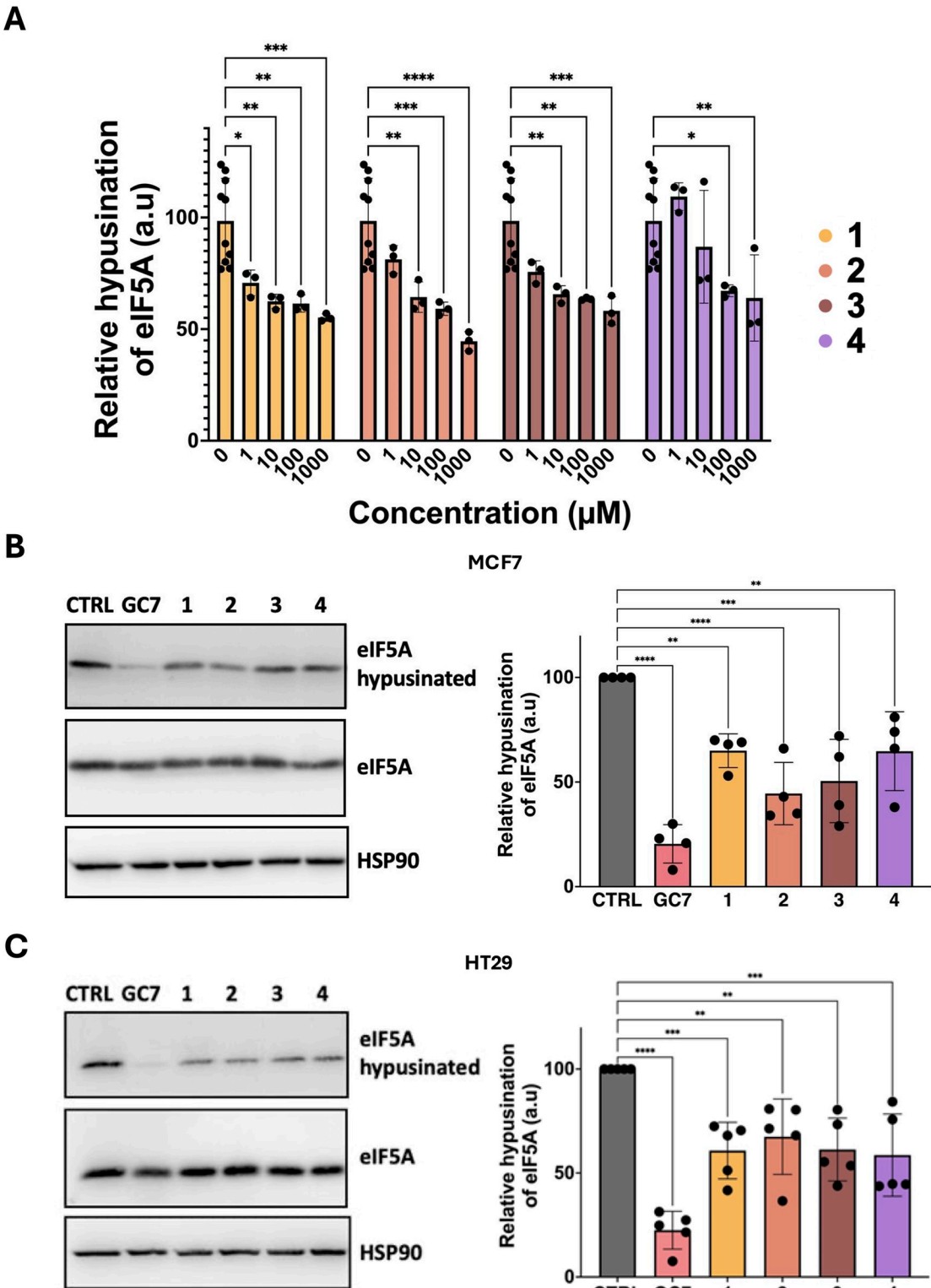

**Fig 5. GC7 analogs are hypusination inhibitors.** (A) Hyp'Assay: Dose-response of GC7 analogs (**1**, **2**, **3** and **4**) from 0 to 1 mM in the presence of 5 μM of spermidine. (B) MCF7 or (C) HT29 were treated with 40 μM of GC7, **1**, **2**, **3** or **4** for 24 h. Cells were lysed and analyzed by Western blot using the indicated antibodies. A quantification of 4 and 5 experiments is provided.

is probably one of the reasons why the efficiency of GC7 in a cell-free system is in the nanomolar range while, in intact cells, it is in the micromolar range. Second, DHPS competitive inhibitors have a polyamine-like structure. Polyamines are polycations that bind to proteins and nucleic acids and are involved in several biological mechanisms, independently of eIF5A activation [31, 32]. Thus, a competitive inhibitor for DHPS may have unwanted effects by affecting other signaling functions of polyamines.

Based on crystallographic studies, a series of allosteric inhibitors that target the NAD binding site of DHPS have been designed such as bromobenzothiophene with an $IC_{50}$ of 60 nM [20] and a 5,6-dihydrothieno [2,3-c]pyridine derivative with an $IC_{50}$ of 9 nM [33]. However, these inhibitors have not been tested in cells. Using molecular dynamics simulations and an indirect assay to measure DHPS activity a 5-(2-methoxyphenoxy)-2-phenylpyrimidin-4-amine derivative has been characterized with an $IC_{50}$ of 14 nM [34] and GL-1 (4,6-dichloro-2-(4-iodophenyl)-5-(2-methoxyphenoxy)pyrimidine) with an $IC_{50}$ of 210 nM [35]. In mice, these molecules inhibit melanoma proliferation *in vitro* and *in vivo*.

DOHH inhibitors are even less numerous than DHPS inhibitors. DOHH is an oxygen and non-heme-iron-dependent enzyme. The only known DOHH inhibitors are iron chelators: ciclopirox, mimosine, and deferiprone [36, 37]. Because of their mode of action, none of them are specific. For instance, ciclopirox is an antifungal drug used since the early 80s that functions by inhibiting several iron-dependent enzymes such as ribonucleotide reductase, and proline hydroxylase [37, 38].

So far, DHPS and DOHH inhibitors have been isolated from educated guesses and/or computational design, followed by the synthesis of a limited number of candidate molecules and an enzymatic assay. Although these approaches can be fruitful (as described above) they are limited in scope. Screening of chemical libraries appears as a more thorough approach. To the best of our knowledge, such a screen has never been undertaken, probably because of the absence of a suitable assay. The results presented here indicated that the Hyp'Assay could fulfill this purpose. Compared to existing methods it could be set up in most of the laboratories. Indeed, besides classical chemicals, it necessitates only purified fusion proteins and an antibody against hypusinated eIF5A. Measures of $EC_{50}$ for spermidine, or $IC_{50}$ for inhibitors using the Hyp'Assay gave similar results to data already published that were obtained using radioactive spermidine, underscoring the reliability of the Hyp'Assay. Another strength of the Hyp'Assay is its ability to find inhibitors for both DHPS and DOHH. Indeed, if DHPS is considered the "priority target" for eIF5A inhibition [13], DOHH is also necessary for eIF5A hypusination and activity. For instance, DOHH is overexpressed in glioblastoma, and its inhibition prolonged survival in mouse models for this disease [39]. A single screen would be sufficient to isolate DHPS and DOHH inhibitors. After finding an inhibitor of hypusination adding them during the first or the second step of the Hyp'Assay, or performing the Hyp'Assay using antibody Hpu98 that recognized deoxyhypusinated-eIF5A, will allow us to determine if the inhibition affects DHPS or DOHH (as described in Fig 3D,3F and 3G). Although, the Hyp'Assay could be a powerful tool to find inhibitors in a cell-free system it cannot predict the efficiency of molecules to inhibit hypusination in intact cells or in whole organism which depend upon their cellular uptake, subcellular distribution and stability. Some chemical libraries, such as the Prestwick or TargetMol library, contained FDA-approved molecules or molecules already described for their bioavailability. For other libraries, or molecules synthesized based on computational design, bioavailability experiments would have to be performed in addition of the Hyp'Assay. Finally, compared to other tests, the Hyp'Assay is affordable for laboratory that can produce fusion proteins, since the only additional costs are negligeable.

Several theoretical mechanisms could lead to an inhibition of hypusination. For instance, inhibition of the association between eIF5A and DHPS as observed for GL-1 [35], or between

DOHH and eIF5A, or inhibition of the tetramerization of DHPS, and probably several other unpredicted manners. Screening of chemical libraries could uncover such new unpredicted inhibitors of hypusination.

Hypusination is also a potential drug target for the treatment of pathogenic parasites such as *T. brucei*, *L. donovani* and *T. vaginalis* [2, 30, 40]. Since the anti hypusine antibodies used in the Hyp'Assay (Hpu24 and Hpu98) are described as independent of the flanking amino acid sequence [18] the Hyp'Assay could be modified by using DHPS, DOHH (when available) and eIF5A from the adequate pathological strain. The research for a specific hypusination inhibitors that affect the parasite and not the host could be difficult since the structure of eIF5A-DHPS complex from human and *T. vaginalis* shows high similarity [30]. However, a modified Hyp'Assay could be a valuable tool to find such inhibitor.

In summary, we present the Hyp'Assay, a powerful and convenient tool that could be used by researchers wanting an easier way to study hypusination and for the discovery of new inhibitors of eIF5A hypusination. This assay could facilitate advancements in the search for novel therapeutic agents that target eIF5A hypusination.

## 4. Material & methods

### 4.1. Reagent and antibodies

CPX, DFX, EDTA, Folate, GC7, Putrescine, Spermidine, and Spermine were bought from Sigma. BENSPM was purchased from MedChemExpress. The antibodies for DHPS (HPA014461) and DOHH (HPA041953) were purchased from Sigma, eIF5A antibody (611977) came from BD Biosciences. Anti Hypusine antibody Hpu24 (PTX18841) and Hpu98 (RGK08103-100) were bought from ProteoGenix. HSP90 antibody (sc-13119) were bought from Santa Cruz. HRP-conjugated AffiniPure donkey anti-mouse IgG (H+L) (711–035152) were purchased from Jackson ImmunoResearch Laboratories.

### 4.2. Cell culture

The human breast cancer cell line MCF7 (HTB-22 ATCC) and colorectal adenocarcinoma HT29 (HTB-38 ATCC) were cultured in DMEM (Gibco) supplemented with 10% FBS (Dutscher), 10 U/mL penicillin, and 10 mg/mL streptomycin. 1 mM of aminoguanidine was added to the medium to inhibit the degradation of GC7 and polyamines by amine oxidase from the serum and the production of reactive oxygen species [41]. Cells were maintained at 37°C, 5% $CO_2$ in a humid atmosphere.

### 4.3. Purification of the fusion proteins DHPS, DOHH, and eIF5A

pET-pelB/6xhis/TEV -eIF5A, -DHPS, and -DOHH (human) were ordered on VectorBuilder. pET/6xhis/TEV-DHPS was produced by deletion of the pelB sequence. pET-pelB/6xhis/TEV-eIF5A and -DOHH and pET /6xhis/TEV-DHPS were expressed in the *E. coli* strain BL21 (DE3). An overnight culture was diluted 1:10 with LB medium (500 mL) until an $OD_{600nm}$ of 0.7 to 0.9 was reached. Proteins were induced with 1 mM IPTG (isopropyl-$\beta$-D-thiogalacto-pyranoside) for 4 h at 37°C. After centrifugation, bacteria were resuspended in 20 mL of Lysis Buffer (100 mM Tris pH 8.0, 250 mM NaCl, 10 mM Imidazole) supplemented with a protease inhibitor cocktail (complete, Roche). This suspension was passed twice through a French Press at $9.10^3$ kPa and centrifuged. The supernatant was incubated with 1 mL of Ni-NTA Agarose beads (Qiagen) for at least 1 h at 4°C and loaded in a polypropylene column. The column was washed once with 5 mL of Lysis Buffer and twice with 5 mL of Wash Buffer (100 mM Tris pH 8.0, 250 mM NaCl, 20 mM Imidazole). Proteins were eluted with 100 mM Tris pH 8.0, 250

mM NaCl, 300 mM Imidazole and dialyzed overnight at 4˚C in Dialysis Buffer (50 mM Tris pH 8.0, 150 mM NaCl) in a SnakeSkin dialysis coil (3 500d). Protein concentration was measured using a BCA assay and integrity was verified by SDS-PAGE followed by Coomassie blue staining.

## 4.4. Hypusination reaction in a cell-free system

eIF5A (5 μg), DHPS (1 μg), and 100 μM spermidine were incubated in 50 μL of 200 μM Tris pH 8.0, 500 μM NAD, 1 mM DTT for 2 h at 37˚C. Then 10 μL containing 1 μg of DOHH with 10 μM $Fe^{2+}$ ($H_8FeN_2O_8S_2 . 6H_2O$) were added. After 1 h proteins were analyzed by Western Blot using antibodies toward DHPS, DOHH, Hypusine, and eIF5A.

## 4.5. Western blot

After treatment, cells were washed with ice-cold PBS and lysed in RIPA Buffer (50 mM Tris pH 7.5, 150 mM NaCl, 0.1% SDS, 0.5% Na Deoxycholate, 5 mM NaF, 2.5 mM $Na_4P_2O_7$, 1% NP4O) containing a protease inhibitor cocktail (complete Roche). 10 μg of proteins were separated by SDS PAGE and transferred to a PVDF membrane (Immobilon-P Millipore). The membranes were probed with the appropriate primary antibody that was detected by horseradish peroxidase-conjugated secondary antibody (Jackson Immunoresearch) and visualized by chemiluminescence (ECL Amersham RPN 3244) using a Fuji LAS-4000 imager. ImageJ® software was used to quantify band intensity and the ratios of proteins of interest were normalized to a loading control. Full-length Western blots are provided in the supplementary data (S19, S20 Figs).

## 4.6. Hyp'Assay

The cell-free reaction described in 4.4 is performed in triplicates in a 96-wells plate (MaxiSorp, ThermoFisher) in the presence of 3 μg of eIF5A, 2 μg of DHPS, and 2 μg of DOHH by well. The reaction product, hypusinated eIF5A, is adsorbed onto the plate overnight at 4˚C. The plate is washed (100 μL/well) thrice with PBS-T (PBS– 0.05% Tween20), then saturated with 1% non-fat dry milk in PBS-T for 2 h at room temperature. The plate is washed thrice with PBS-T and incubated overnight at 4˚C with the anti-hypusine antibody (1:2000 in saturation buffer, 50 μL/well). The next day and after 5 washes with PBS-T, HRP-coupled anti-rabbit secondary antibody diluted 1:5000 in saturation buffer (50 μL/well) is added for 2 h at room temperature. After 5 washes, the plate is revealed using TMB (Tetramethylbenzidine (Applied Biological Materials) as a substrate (100 μL/well) in the dark for 10 minutes. The reaction is stopped by the addition of 2N $H_2SO_4$ (100 μL/well). Absorbance is measured with a Multiskan™ FC photometer at 450nm. Controls in the absence of spermidine (blank) are performed.

## 4.7. Statistical analysis

Statistical tests were performed using GraphPad Prism software. Comparisons of conditions are analyzed with a One-way ANOVA or a Two-way ANOVA test. All our hypotheses are established with an alpha of 0.05%. For our p-values or adjusted-p-values, we use the following legend: ns: p-value > 0.05; *: p-value ≤ 0.05; **: p-value ≤ 0.01; ***: p-value ≤ 0.001; **** p-value ≤ 0.0001.

## 4.8. GC7 analogs synthesis

All solvents and reagents for synthesis used are ACS Reagent minimum quality. [1]H and [13]C NMR analyses were carried out using Bruker Avance 400 MHz (Brüker, Rheinstetten,

Germany). Chemical shifts were reported in delta units ($\delta$) parts per million (ppm) downstream of tetramethylsilane (TMS) as internal standard. The deuterated solvents used to solubilize the compounds were used to calibrate proton (thanks to the residual solvent signal) and carbon signals (Chloroform-$d$: $\delta_H$ = 7.26 ppm and $\delta_C$ = 77.0 ppm; methanol-$d_4$: $\delta_H$ = 3.31 ppm and $\delta_C$ = 49.0 ppm) Peak configurations are indicated as follows: $s$, singlet; $d$, doublet; $t$, triplet; $m$, multiplet; $q$, quartet. Coupling constants ($J$) are given in Hertz (Hz). Flash column chromatography was performed on a Puriflash XS 520 Plus system with silica gel columns (Interchim Puriflash silica SI, 50 $\mu$m). The gradient used for the different purifications via the flash chromatography column was optimized to ensure that the separation of compounds was adapted to each reaction. NMR spectra are provided in S3–S18 Figs.

**General protocol A**: Cleavage of Boc protecting groups

The various guanidine chains protected by the Boc groups (1.0 eq.) are solubilized in a minimum of $CH_2Cl_2$, then a 50% solution of TFA in $CH_2Cl_2$ (0.2 M) is added. The solution is stirred at room temperature for 2 h. The $CH_2Cl_2$ is evaporated under reduced pressure. Excess TFA is removed by co-evaporation with MeOH and freeze-drying. The yields obtained are quantitative.

### 4-hydroxybutylguanidine (1)

Synthetized according to the **general protocol A**. White solid, TFA salt; HRESI(+)MS $m/z$ 132.1137 $[M + H]^+$ (calcd for $C_5H_{14}N_3O^+$, 132.1131); $^1$H NMR (400 MHz, methanol-$d_4$), $\delta$ppm (mult., $J$(Hz)): 4.43 ($t$, $^3J$ = 6.4 Hz, 2H, H-2), 3.23 ($t$, $^3J$ = 7.0 Hz, 2H, H-5), 1.90–1.73 ($m$, 2H, H-4), 1.73–1.62 ($m$, 2H, H-3); $^{13}$C NMR (101 MHz, methanol-$d_4$), $\delta$ppm: 157.3 ($C_q$-1), 60.9 ($CH_2$-5), 40.9 ($CH_2$-2), 29.0 ($CH_2$-3), 25.2 ($CH_2$-4).

### 6-hydroxyhexyguanidine (2)

Synthetized according to the **general protocol A**. White solid, TFA salt; HRESI(+)MS $m/z$ 160.1424 $[M + H]^+$ (calcd for $C_7H_{18}N_3O^+$, 160.1444); $^1$H NMR (400 MHz, methanol-$d_4$), $\delta$ppm (mult., $J$(Hz)): 4.40 ($t$, $^3J$ = 6.5 Hz, 2H, H-7), 3.18 ($t$, $^3J$ = 7.1 Hz, 2H, H-2), 1.86–1.72 ($m$, 2H, H-6), 1.68–1.49 ($m$, 2H, H-3), 1.51–1.36 ($m$, 4H, H-4, H-5); $^{13}$C NMR (101 MHz, methanol-$d_4$), $\delta$ppm: 158.7 ($C_q$-1), 62.8 ($CH_2$-7), 42.4 ($CH_2$-2), 33.4 ($CH_2$-6), 29.8($CH_2$-3), 27.5 ($CH_2$-4), 26.5 ($CH_2$-5).

### 7-hydroxyheptylguanidine (3)

Synthetized according to the **general protocol A**. White solid, TFA salt; HRESI(+)MS $m/z$ 174.1603 $[M + H]^+$ (calcd for $C_8H_{20}N_3O^+$, 174.1601); $^1$H NMR (400 MHz, methanol-$d_4$), $\delta$ppm (mult., $J$(Hz)): 4.38 ($t$, $^3J$ = 6.6 Hz, 2H, H-8), 3.17 ($t$, $^3J$ = 7.1 Hz, 2H, H-2), 1.76 ($t$, $^3J$ = 7.0 Hz, 2H, H-7), 1.63–1.55 ($m$, 2H, H-3), 1.45–1.36 ($m$, 6H, H-4, H-5, H-6); $^{13}$C NMR (101 MHz, methanol-$d_4$), $\delta$ppm: 158.7 ($C_q$-1), 69.5 ($CH_2$-8), 42.4 ($CH_2$-2), 33.5 ($CH_2$-6), 30.1 ($CH_2$-3), 29.8 ($CH_2$-8), 27.6 ($CH_2$-4), 26.8 ($CH_2$-5).

### 8-hydroxyoctylguanidine (4)

Synthetized according to the **general protocol A**. White solid, TFA salt; HRESI(+)MS $m/z$ 188.1733 $[M + H]^+$ (calcd for $C_9H_{22}N_3O^+$, 188.1757); $^1$H NMR (400 MHz, methanol-$d_4$), $\delta$ppm (mult., $J$(Hz)): 4.39 ($t$, $^3J$ = 6.6 Hz, 2H, H-2), 3.16 ($t$, $^3J$ = 7.1 Hz, 2H, H-9), 1.76 ($t$, $^3J$ = 7.0 Hz, 2H, H-3), 1.58 ($d$, $^3J$ = 7.0 Hz, 2H, H-8), 1.41–1.33 ($m$, 8H, H-4, H-5, H-6, H-7); $^{13}$C NMR (101 MHz, methanol-$d_4$), $\delta$ppm: 157.3 ($C_q$-1), 68.1 ($CH_2$-9), 41.0 ($CH_2$-2), 28.7 ($CH_2$-8), 28.6 ($CH_2$-3), 27.8 ($CH_2$-6), 26.2 ($CH_2$-7), 26.1 ($CH_2$-5).

**General protocol B:** Synthesis of the Boc protect guanidine derivatives

The corresponding aminoalcohol (1.5 eq.) and 1.2 eq. of DIPEA are introduced into MeOH (0.25 M) at room temperature. To the reaction mixture are added 1.0 eq. of 1$H$-pyrazole-1-($N$, $N$'-di-$tert$-butoxycarbonyl)carboxamidine (**6**). The mixture is stirred at room temperature for 5 h. The solvent is evaporated under reduced pressure and the residue is taken up in EtOAc. The organic phase is washed with a solution of $KHSO_4$ (1.0 M) followed by distilled water. The organic phase is dried over $MgSO_4$ and, after filtration, evaporated under reduced pressure and purified.

### *N,N*'-di-*tert*-butoxycarbonyl-4-hydroxybutylguanidine (5)

According to **general protocol B**, 4-aminobutanol (108 mg, 1.21 mmol, 1.5 eq.), DIPEA (124 mg, 0.96 mmol, 1.2 eq.) and 1H-pyrazole-1-(N,N'-di-tert-butoxycarbonyl)carboxamidine (6) (250 mg, 0.80 mmol, 1.0 eq.) is stirred at room temperature for 5 h. After work-up, a purification on flash chromatography was done with cyclohexane/EtOAc, 8:2, v/v; the compound was obtained pure (110 mg, 42%).White solid; HRESI(+)MS $m/z$ 332.2190 [M + H]$^+$ (calcd for $C_{15}H_{30}N_3O_5^+$, 332.2179); $^1$H NMR (400 MHz, methanol-$d_4$), δppm (mult., $J$(Hz)): 4.71 (s, 3H, H-6, H-7, H-8), 3.49 (t, $^3J$ = 6.2 Hz, 2H, H-5), 3.29 (t, $^3J$ = 6.8 Hz, 2H, H-2), 1.60–1.46 (m, 4H, H-3, H-4), 1.42 (s, 9H, H-11, H-12, H-13), 1.37 (s, 9H, H-11', H-12', H-13'); $^{13}$C NMR (101 MHz, methanol-$d_4$), δppm: 163.2 (C$_q$-1), 156.2 (C$_q$-9), 152.8 (C$_q$-9'), 83.0 (C$_q$-10), 78.9 (C$_q$-10'), 61.1 (CH$_2$-5), 40.2 (CH$_2$-2), 29.4 (CH$_2$-3), 27.3 (CH$_3$-11, CH$_3$-12, CH$_3$-13), 27.0 (CH$_3$-11', CH$_3$-12', CH$_3$-13'), 25.3 (CH$_2$-4).

### *N,N*'-di-*tert*-butoxycarbonyl-6-hydroxyhexylguanidine (6)

According to **general protocol B**, 6-aminohexanol (141 mg, 1.21 mmol, 1.5 eq.), DIPEA (124 mg, 0.96 mmol, 1.2 eq.) and 1$H$-pyrazole-1-($N,N$'-di-$tert$-butoxycarbonyl)carboxamidine (250 mg, 0.805 mmol, 1.0 eq.) is stirred at room temperature for 5 h. After work-up, a purification on flash chromatography was done with Cyclohexane/EtOAc, 8:2, v/v; the compound was obtained pure (67 mg, 53%). White solid; HRESI(+)MS $m/z$ 360.2496 [M + H]$^+$ (calcd for $C_{17}H_{34}N_3O_5^+$, 360.2492); $^1$H NMR (400 MHz, methanol-$d_4$), δppm (mult., $J$(Hz)): 3.55 (t, $^3J$ = 6.6 Hz, 2H, H-2), 3.36 (t, $^3J$ = 7.1 Hz, 2H, H-7), 1.66–1.55 (m, 4H, H-6, H-3), 1.53 (s, 9H, H-13, H-14, H-15), 1.47 (s, 9H, H-13', H-14', H-15'), 1.42–1.30 (m, 4H, H-4, H-5); $^{13}$C NMR (101 MHz, methanol-$d_4$), δppm: 163.2 (C$_q$-1), 156.2 (C$_q$-11), 152.9 (C$_q$-11'), 104.2 (C$_q$-12'), 83.1 (C$_q$-12), 61.5 (CH$_2$-7), 40.4 (CH$_2$-2), 32.1 (CH$_2$-6), 28.7 (CH$_2$-3), 27.2 (CH$_3$-13, CH$_3$-14, CH$_3$-15), 26.8 (CH$_3$-13', CH$_3$-14', CH$_3$-15'), 26.3 (CH$_2$-4), 25.2 (CH$_2$-5).

### *N,N*'-di-*tert*-butoxycarbonyl-7-hydroxyheptylguanidine (7)

According to **general protocol B**, 7-aminoheptanol (158 mg, 1.21 mmol, 1.5 eq.), DIPEA (124 mg, 0.96 mmol, 1.2 eq.) and 1$H$-pyrazole-1-($N,N$'-di-$tert$-butoxycarbonyl)carboxamidine (250 mg, 0.80 mmol, 1.0 eq) is stirred at room temperature for 5 h. After work-up, a purification on flash chromatography was done with Cyclohexane/EtOAc, 7:3, v/v; the compound was obtained pure (77 mg, 56%). White solid; HRESI(+)MS $m/z$ 374.2679 [M + H]$^+$ (calcd for $C_{18}H_{36}N_3O_5^+$, 374.2649); $^1$H NMR (400 MHz, methanol-$d_4$), δppm (mult., $J$(Hz)): 4.85 (s, 3H, H-9, H-10, H-11), 3.57 (t, $^3J$ = 6.6 Hz, 2H, H-2), 3.37 (t, $^3J$ = 7.1 Hz, 1H, H-8), 1.60 (m, 4H, H-3, H-7), 1.55 (s, 9H, H-14, H-15, H-16), 1.49 (s, 9H, H-14', H-15', H-16'), 1.40 (s, 6H, H-4, H-5, H-6); $^{13}$C NMR (101 MHz, methanol-$d_4$), δppm: 163.2 (C$_q$-1), 156.2 (C$_q$-12), 152.9 (C$_q$-12'), 83.1 (C$_q$-13), 78.9 (C$_q$-13'), 61.5 (CH$_2$-8), 40.4 (CH$_2$-2), 32.2 (CH$_2$-7), 28, 8 (CH$_2$-5), 28.6 (CH$_2$-3), 27.2 (CH$_3$-14, CH$_3$-15, CH$_3$-16), 26.8 (CH$_3$-14', CH$_3$-15', CH$_3$-16'), 26.5 (CH$_2$-4), 25.4 (CH$_2$-6).

### N,N'-di-tert-butoxycarbonyl-8-hydroxyloctylguanidine (8)

According to **general protocol B**, 8-aminooctanol (175 mg, 1.2 mmol, 1.5 eq.), DIPEA (124 mg, 0.96 mmol, 1.2 eq.) and 1H-pyrazole-1-(N,N'-di-tert-butoxycarbonyl)carboxamidine (250 mg, 0.80 mmol, 1.0 eq.) is stirred at room temperature for 5 h. After work-up, a purification on flash chromatography was done with Cyclohexane/EtOAc, 8:2, v/v; the compound was obtained pure (82 mg, 55%). White solid; HRESI(+)MS $m/z$ 388.2733 [M + H]$^+$ (calcd for $C_{19}H_{38}N_3O_5^+$, 388.2805); $^1$H NMR (400 MHz, methanol-$d_4$), δppm (mult., J(Hz)): 3.54 ($t$, $^3J$ = 6.6 Hz, 2H, H-9), 3.35 ($t$, $^3J$ = 7.1 Hz, 2H, H-2), 1.58 ($t$, $^3J$ = 7.0 Hz, 4H, H-3, H-8), 1.53 ($s$, 9H, H-15, H-16, H-17), 1.47 ($s$, 9H, H-15', H-16', H-17'), 1.38–1.26 ($m$, 8H, H-4, H-5, H-6, H-7); $^{13}$C NMR (101 MHz, methanol-$d_4$), δppm: 164.6 ($C_q$-1), 157.6 ($C_q$-13), 154.3 ($C_q$-13'), 84.5 ($C_q$-14), 80.3, ($C_q$-14'), 63.0 ($CH_2$-9), 41.8($CH_2$-2), 33.6 ($CH_2$-8), 30.4 ($CH_2$-5), 30.3 ($CH_2$-6), 30.1 ($CH_2$-3), 28.6 ($CH_3$-15, $CH_3$-16, $CH_3$-17), 28.2 ($CH_3$-15', $CH_3$-16', $CH_3$-17'), 27.8 ($CH_2$-4), 26.8 ($CH_2$-7).

## Supporting information

**S1 Fig. Bacterial construction of recombinant human eIF5A, DHPS or DOHH proteins and corresponding elution profile of column-purified proteins analysed by SDS-PAGE and stained with Coomassie blue.**
(DOCX)

**S2 Fig. Reaction of hypusination of eIF5A performed at various pH.**
(DOCX)

**S3 Fig. Reaction of hypusination of eIF5A performed at various concentration of spermidine or spermine.**
(DOCX)

**S4 Fig. 1H NMR spectrum (400 MHz, methanol-d4) of N,N'-di-tert-butoxycarbonyl-4-hydroxybutylguanidine (5).**
(DOCX)

**S5 Fig. 13C NMR spectrum (101 MHz, methanol-d4) of N,N'-di-tert-butoxycarbonyl-4-hydroxybutylguanidine (5).**
(DOCX)

**S6 Fig. 1H NMR spectrum (400 MHz, methanol-d4) of 4-hydroxybutylguanidine (1).**
(DOCX)

**S7 Fig. 13C NMR spectrum (101 MHz, methanol-d4) of 4-hydroxybutylguanidine (1).**
(DOCX)

**S8 Fig. 1H NMR spectrum (400 MHz, methanol-d4) of N,N'-di-tert-butoxycarbonyl-6-hydroxyhexylguanidine (6).**
(DOCX)

**S9 Fig. 13C NMR spectrum (101 MHz, methanol-d4) of N,N'-di-tert-butoxycarbonyl-6-hydroxyhexylguanidine (6).**
(DOCX)

**S10 Fig. 1H NMR spectrum (400 MHz, methanol-d4) of 6-hydroxyhexylguanidine (2).**
(DOCX)

**S11 Fig. 13C NMR spectrum (101 MHz, methanol-d4) of 6-hydroxyhexylguanidine (2).**
(DOCX)

**S12 Fig. 1H NMR spectrum (400 MHz, methanol-d4) of N,N'-di-tert-butoxycarbonyl-7-hydroxyheptylguanidine (7).**
(DOCX)

**S13 Fig. 13C NMR spectrum (101 MHz, methanol-d4) of N,N'-di-tert-butoxycarbonyl-7-hydroxyheptylguanidine (7).**
(DOCX)

**S14 Fig. 1H NMR spectrum (400 MHz, methanol-d4) of 7-hydroxyheptylguanidine (3).**
(DOCX)

**S15 Fig. 13C NMR spectrum (101 MHz, methanol-d4) of 7-hydroxyheptylguanidine (3).**
(DOCX)

**S16 Fig. 1H NMR spectrum (400 MHz, methanol-d4) of N,N'-di-tert-butoxycarbonyl-8-hydroxyoctylguanidine (8).**
(DOCX)

**S17 Fig. 13C NMR spectrum (101 MHz, methanol-d4) of N,N'-di-tert-butoxycarbonyl-8-hydroxyoctylguanidine (8).**
(DOCX)

**S18 Fig. 1H NMR spectrum (400 MHz, methanol-d4) of 8-hydroxyoctylguanidine (4).**
(DOCX)

**S19 Fig. 13C NMR spectrum (101 MHz, methanol-d4) of 8-hydroxyoctylguanidine (4).**
(DOCX)

**S20 Raw images.**
(PDF)

## Acknowledgments

We want to thank researchers from team 6 of C3M for their help with the handling of bacteria and the French Press and Dr. Sophie Giorgetti-Peraldi for helpful discussions.

## Author Contributions

**Conceptualization:** Mohamed Mehiri, Frederic Bost, Pascal Peraldi.

**Data curation:** Pascal Peraldi.

**Funding acquisition:** Frederic Bost.

**Investigation:** Oumayma Benaceur, Paula Ferreira Montenegro, Michel Kahi, Fabien Fontaine-Vive, Nathalie M. Mazure, Pascal Peraldi.

**Supervision:** Pascal Peraldi.

**Writing – original draft:** Oumayma Benaceur, Paula Ferreira Montenegro, Mohamed Mehiri, Frederic Bost, Pascal Peraldi.

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
