## [Decision Letter · Decision Letter 0]

6 Sep 2024

PONE-D-24-29255Development of a reliable, sensitive, and convenient assay

 for the discovery of new eIF5A hypusination inhibitorsPLOS ONE

Dear Dr. Peraldi,

Thank you for submitting your manuscript to PLOS ONE. After careful consideration, we feel that it has merit but does not fully meet PLOS ONE’s publication criteria as it currently stands. Therefore, we invite you to submit a revised version of the manuscript that addresses the points raised during the review process.

We look forward to receiving your revised manuscript.

Kind regards,

Jorddy Neves Cruz

Academic Editor

PLOS ONE

Journal requirements: 1. When submitting your revision, we need you to address these additional requirements. Please ensure that your manuscript meets PLOS ONE's style requirements, including those for file naming. The PLOS ONE style templates can be found at https://journals.plos.org/plosone/s/file?id=wjVg/PLOSOne_formatting_sample_main_body.pdf and https://journals.plos.org/plosone/s/file?id=ba62/PLOSOne_formatting_sample_title_authors_affiliations.pdf. 2. PLOS ONE now requires that authors provide the original uncropped and unadjusted images underlying all blot or gel results reported in a submission’s figures or Supporting Information files. This policy and the journal’s other requirements for blot/gel reporting and figure preparation are described in detail at https://journals.plos.org/plosone/s/figures#loc-blot-and-gel-reporting-requirements and https://journals.plos.org/plosone/s/figures#loc-preparing-figures-from-image-files. When you submit your revised manuscript, please ensure that your figures adhere fully to these guidelines and provide the original underlying images for all blot or gel data reported in your submission. See the following link for instructions on providing the original image data: https://journals.plos.org/plosone/s/figures#loc-original-images-for-blots-and-gels.   In your cover letter, please note whether your blot/gel image data are in Supporting Information or posted at a public data repository, provide the repository URL if relevant, and provide specific details as to which raw blot/gel images, if any, are not available. Email us at plosone@plos.org if you have any questions. 3. We note that the grant information you provided in the ‘Funding Information’ and ‘Financial Disclosure’ sections do not match.  When you resubmit, please ensure that you provide the correct grant numbers for the awards you received for your study in the ‘Funding Information’ section.

Reviewers' comments:

Reviewer's Responses to Questions

**Comments to the Author**

1. Is the manuscript technically sound, and do the data support the conclusions?

Reviewer #1: Yes

Reviewer #2: Yes

Reviewer #3: Yes

Reviewer #4: Yes

2. Has the statistical analysis been performed appropriately and rigorously? 

Reviewer #1: Yes

Reviewer #2: Yes

Reviewer #3: N/A

Reviewer #4: Yes

3. Have the authors made all data underlying the findings in their manuscript fully available?

Reviewer #1: Yes

Reviewer #2: Yes

Reviewer #3: Yes

Reviewer #4: Yes

4. Is the manuscript presented in an intelligible fashion and written in standard English?

Reviewer #1: Yes

Reviewer #2: Yes

Reviewer #3: Yes

Reviewer #4: Yes

5. Review Comments to the Author

Reviewer #1: Benaceur et al. ”Development of a reliable, sensitive, and convenient assay for the discovery of new eIF5A hypusination inhibitors”

The manuscript describes an in vitro method to quantitate eIF5A hypusination using purified recombinant protein, anti-hypusinated eIF5A antibodies and western blot. Both proteins required for the hypusination, deoxyhypusine synthase (DHPS) and deoxyhypusine hydroxylase (DOHH) are present in the in vitro reaction, thus both reactions can be assayed separately – this is a clear advantage of the system. The established DHPS inhibitor, GC7, several polyamines, as well as new DHPS inhibitors designed in this work, were tested for their ability to inhibit eIF5A hypusination in this system.

A major question is how much is new information in this manuscript. A similar method was recently published (Wator et al. Nature Comm 14:1698, 2023). This paper is not cited by the authors but I think it should be. With this in mind, it could be considered to change the word “development” in the title to “adaptation” or similar.

What actually differs from the Wator paper is that new tentative DHPS inhibitors were developed and tested, and that the two reactions can be assayed separately using the two enzymes. The assay was also adapted for 96 well format. The antibody used in this work detects hypusine, whereas the antibody used in Wator et al. detects both hypusine and deoxyhypusine. It should be considered how the present method is different from Wator et al., and discussed if it is superior in some respects.

Regarding the utility of the method: if it is to be scaled up to screen large libraries, as proposed by the authors, what would be the cost given that 1 µg of each enzyme is needed per reaction?

The new DHPS inhibitors developed in this work were found not to very effective. For all of these inhibitors, only partial inhibition (~50%) of hypusination was achieved in vitro, even at very high concentrations (1 mM) in the presence of 5 µM spermidine (Fig. 5), making IC50 values just barely possible to calculate. By contrast, under the same conditions GC7 achieves ~95% inhibition at 1 µM, giving an IC50 of 6.8 nM (Fig. 3). Why does the inhibition with the new compounds not go higher?

In vivo (cultured human cells), the new molecules were tested at 40 µM and achieve partial inhibition there (~50%) (Fig. 5 B). The authors also discuss (line 312 onwards) DHPS inhibitors developed by other groups, with nM IC50 values in vitro. Some of these have not been tested in intact cells; in one case in vivo effects have been seen but the concentrations needed are not mentioned. The authors should discuss the challenges of developing inhibitors based on cell-free testing systems, where cellular uptake, subcellular distribution and biostability are not taken into account.

Minor comments:

Line 41: “carbon 9 of deoxy-hypusinated eIF5A” should be “carbon 9 of Lys50 of deoxy-

hypusinated eIF5A”

Line 184: How do the authors mean if they want to exclude that the inhibition by GC7 was due to a non-specific polyamine effect? After all, GC7 is a spermidine analog so should not such an effect be expected? Moreover, how do the tests with other polyamines exclude or include the possibility that the inhibition by GC7 is a non-specific polyamine effect?

Reviewer #2: The manuscript by Benaceur et al presents a development of the assay capable of measuring the hypusination activity with the potential to be used as a tool to discover new hypusination inhibitors, DHPS or DOHH inhibitors. As the Authors stated, the reliable, easy readout assay of hypusination is missing ad as such the study by Benaceur et al answers the demand for developing such a method. Overall, I find the data in the manuscript scientifically sound and would like to support the publication in PLOS One after major revision and addressing the following comments.

1. The Authors used Hpu24 clone of the Hpu-specific antibody. This implies the detection of the fully hypusinated eIF5a. However, there is an available anti-Hpu/Dhp antibody (Hpu98) that detects both, hypusinated and deoxyhypusinated eIF5a. Using Hpu98 instead of Hpu24 would allow to screen only for DHPS inhibitors and the omission of DOHH from the assay. The authors should consider evaluating such a modification for their assay. Especially that it would go in line with the authors' statement in Line 345 “Indeed, if DHPS is considered the 346 “priority target” for eIF5A inhibition” and ib Line 209 “Although these experiments show that the Hyp’Assay could be used to find new hypusination inhibitors, as it is, it cannot discriminate between DHPS and DOHH inhibitors.”

2. As mentioned above there are Hpu24 and Hpu98 antibodies available. The citation of the study describing them must be included 10.1016/j.jmb.2016.01.006

3. The optimal pH for the assay was tested and found to be 8.0. The optimal pH for DHPS-catalyzed reaction seems to be above 9.2 (Park et al). In the case of using Hpu98 and using Hyp’assay to assess the DHPS activity it is necessary to check pH dependency

4. Authors noticed that putrescine but not spermine has an inhibitory effect. Spermine is the substrate for the oxidoreductase reaction catalyzed by DHPS. As such, it may be possible that spermine is a donor of aminobutyl moiety to deoxyhypusinate eIF5a. Authors should challenge this using spermine as a substrate in Hyp’assay.

5. Recently another study employed hypusine-specific antibodies to dectect hypusination (10.1111/febs.17207) in parasites. Authors should relate to this study. Also, this raises the possibility of using Hyp’assay to detect inhibitors of the hypusination in parasites. There are several attempts to develop such inhibitors in B. malay, L. donovani or other organisms such as tick. The authors should at least discuss such a possibility for their assay.

6. In the introduction some important citations are missing 10.1016/j.str.2024.03.008 and 10.1038/s41467-023-37305-2 – studies describing the structural basis of deoxyhypusination in archaea and humans.

Reviewer #3: The manuscript Development of a reliable, sensitive, and convenient assay for the discovery of new eIF5A hypusination inhibitors is an excellent attempt by the Peraldi et al and the results obtained are promising and can be very much beneficial for the readers of the journal, the authors have four new GC7 analogs and characterized them through various techniques and prove the structural assignments through these studies. The overall quality of the manuscript is nice and can be a good addition in the recent available literature of the field. However there are some old/irrelevant citations which should be replace, after careful reading I suggest the following replacements

replace reference number 5, 6 and 7 with

doi: https://doi.org/10.1016/j.ejso.2020.02.034

https://doi.org/10.1007/s00604-023-05652-y

https://doi.org/10.1021/acsaelm.4c00941

With this minor change I recommend this manuscript for publication

Reviewer #4: The authors presented a convenient cell-free assay, Hyp’Assay, to monitor the hypusination of the translation factor eIF5A, and they performed screening of four GC7 analogs as new hypusination inhibitors they synthesized. It is interesting. However, I am confused with a major issue. It is suggested that the blockade of eIF5A hypusination limited cancerous cell growth in colorectal cancer, which needs further validation in the breast cancer. Why did the authors select a breast cancer cell line MCF-7, but not a classical colorectal cancer cell line, to test the reliability of the Hyp’Assay? The test with human colon cancer cell lines should be performed.

6. PLOS authors have the option to publish the peer review history of their article (what does this mean?). If published, this will include your full peer review and any attached files.

Reviewer #1: No

Reviewer #2: No

Reviewer #3: No

Reviewer #4: **Yes: **Chen-Jie Fang

---

## [Author Response · Author response to Decision Letter 0]

1 Oct 2024

Reviewer #1

Benaceur et al. ”Development of a reliable, sensitive, and convenient assay for the discovery of new eIF5A hypusination inhibitors”

The manuscript describes an in vitro method to quantitate eIF5A hypusination using purified recombinant protein, anti-hypusinated eIF5A antibodies and western blot. Both proteins required for the hypusination, deoxyhypusine synthase (DHPS) and deoxyhypusine hydroxylase (DOHH) are present in the in vitro reaction, thus both reactions can be assayed separately – this is a clear advantage of the system. The established DHPS inhibitor, GC7, several polyamines, as well as new DHPS inhibitors designed in this work, were tested for their ability to inhibit eIF5A hypusination in this system.

A major question is how much is new information in this manuscript. A similar method was recently published (Wator et al. Nature Comm 14:1698, 2023). This paper is not cited by the authors but I think it should be. With this in mind, it could be considered to change the word “development” in the title to “adaptation” or similar. What actually differs from the Wator paper is that new tentative DHPS inhibitors were developed and tested, and that the two reactions can be assayed separately using the two enzymes. The assay was also adapted for 96 well format. The antibody used in this work detects hypusine, whereas the antibody used in Wator et al. detects both hypusine and deoxyhypusine. It should be considered how the present method is different from Wator et al., and discussed if it is superior in some respects.

We would like to thank the reviewer for her/his comments and this remark. 

In the manuscript cited by the reviewer, Wator et al (from Dr. Grudnik lab) measured the catalytic activity of DHPS mutants using a modification of the NAD assay from Park et al (ref 13 of our original manuscript.). Then, they performed an anti-hypusine Western blot. As written by the authors, “ To further validate the importance of the above-mentioned residues in the catalytic reaction, we performed a qualitative hypusination assay to directly measure modification activity.”. So, they performed Western blot using anti hypusine antibodies to verify qualitatively the results obtained using the NAD assay. We now discuss the method described by Wator er al. in our manuscript (lines 89-91 ref. 7).

We feel however that our assay is more than a mere adaptation of Wator’s work. Indeed, as pointed by the reviewer, the differences between the Hyp’Assay and the assay used by Wator et al. are:

1-A quantitative assay on 96 wells vs a qualitative assay performed by Western blot (limiting the number of testable conditions). 

2-An assay designed for DHPS and DOHH activities vs an assay designed only for DHPS.

We had the opportunity to present our data at the conference “ Biological roles of polyamines, 7th Yamada Symposium” in August 2024 and several colleagues were interested to establish a collaboration to test potential hypusination inhibitors using the Hyp’Assay.

The use of an antibody that detects both hypusine and deoxyhypusine in our Hyp’Assay is a very interesting question, that has also been raised by reviewer #2. As described in the detailed response to reviewer #2, we have adapted the Hyp’assay for this antibody. This gives us a new tool to study hypusination and reinforces the Hyp’Assay (new figure 3E, 3F, 3G and lines 246-263).

The manuscript of Wator et al. on the structure study of the interaction between DHPS-eIF5A is indeed a landmark paper that we forgot to add in our manuscript. This mistake has been corrected and a paragraph on the eIF5A – DHPS complex has been added in our manuscript (line 40-49, reference 7).

Regarding the utility of the method: if it is to be scaled up to screen large libraries, as proposed by the authors, what would be the cost given that 1 µg of each enzyme is needed per reaction?

 If one had to buy the necessary fusion proteins (eIF5A, DHPS and DOHH) to perform a screen using the Hyp’Assay the price will indeed be very high. However, fusion proteins can be easily produced in-house, as we did. For a library of 10 000 molecules, around 10 mg of each protein will be needed. We produce about 2 mg of purified protein by liter of bacteria in LB medium so we’ll need 5 liters by construct. 15 liters of LB cost around 30€. 25 ml of Ni-NTA agarose costs around 200€ and can be recycled. ELISA plates are about 2€ each. We will need less than 250 µg anti hypusine antibodies (400€ when bought in bulk) and the dilution can be re-used several times. We’ll need around 500 ml of TMB (300€). So the price for screening a 10 000 compounds chemical library is around 1500€ / $ which seems reasonable. The main cost will be labor-related. We have added a sentence concerning the low cost of the Hyp’Assay (line 410-413).

The new DHPS inhibitors developed in this work were found not to very effective. For all of these inhibitors, only partial inhibition (~50%) of hypusination was achieved in vitro, even at very high concentrations (1 mM) in the presence of 5 µM spermidine (Fig. 5), making IC50 values just barely possible to calculate. By contrast, under the same conditions GC7 achieves ~95% inhibition at 1 µM, giving an IC50 of 6.8 nM (Fig. 3). Why does the inhibition with the new compounds not go higher?

 GC7 is a competitive inhibitor of spermidine for DHPS. Compared to GC7, our analogs have different chains lengths and an -OH instead of a -NH2 (Fig. 4A). It is likely that this modification decreases the affinity of our analogs to DHPS compared to GC7 making them less potent inhibitors. In their manuscript describing GC7, Lee et al. screened for DHPS inhibitors and found several molecules with an IC50 superior to 1 mM (ref 26 in our manuscript). A sentence has been added lines 313-315.

In vivo (cultured human cells), the new molecules were tested at 40 µM and achieve partial inhibition there (~50%) (Fig. 5 B). The authors also discuss (line 312 onwards) DHPS inhibitors developed by other groups, with nM IC50 values in vitro. Some of these have not been tested in intact cells; in one case in vivo effects have been seen but the concentrations needed are not mentioned. The authors should discuss the challenges of developing inhibitors based on cell-free testing systems, where cellular uptake, subcellular distribution and biostability are not taken into account.

 This is a good point, and we have added a paragraph in our new version of our manuscript (lines 402-410). 

Minor comments : 

Line 41: “carbon 9 of deoxy-hypusinated eIF5A” should be “carbon 9 of Lys50 of deoxy-

hypusinated eIF5A”

This has been corrected it in the new version of our manuscript.

Line 184: How do the authors mean if they want to exclude that the inhibition by GC7 was due to a non-specific polyamine effect? After all, GC7 is a spermidine analog so should not such an effect be expected? Moreover, how do the tests with other polyamines exclude or include the possibility that the inhibition by GC7 is a non-specific polyamine effect?

 We agree with the reviewer, this sentence was poorly written. What we meant was that GC7, with its carbon chain and its charged groups, is an amphiphilic molecule. Some reviewers have suggested that some of its effects could be the result of its detergent-like properties. To address this, we tested the ability of other polyamines, which are amphiphilic like GC7, to inhibit hypusination and found that they were ineffective to inhibit hypusination, ruling out a detergent effect of GC7. Since this sentence was misleading, we deleted it (lines 209-210 of the original manuscript)

Reviewer #2.

The manuscript by Benaceur et al presents a development of the assay capable of measuring the hypusination activity with the potential to be used as a tool to discover new hypusination inhibitors, DHPS or DOHH inhibitors. As the Authors stated, the reliable, easy readout assay of hypusination is missing ad as such the study by Benaceur et al answers the demand for developing such a method. Overall, I find the data in the manuscript scientifically sound and would like to support the publication in PLOS One after major revision and addressing the following comments.

We would like to thank the reviewer for her/his kind comments.

The Authors used Hpu24 clone of the Hpu-specific antibody. This implies the detection of the fully hypusinated eIF5a. However, there is an available anti-Hpu/Dhp antibody (Hpu98) that detects both, hypusinated and deoxyhypusinated eIF5a. Using Hpu98 instead of Hpu24 would allow to screen only for DHPS inhibitors and the omission of DOHH from the assay. The authors should consider evaluating such a modification for their assay. Especially that it would go in line with the authors' statement in Line 345 “Indeed, if DHPS is considered the 346 “priority target” for eIF5A inhibition” and ib Line 209 “Although these experiments show that the Hyp’Assay could be used to find new hypusination inhibitors, as it is, it cannot discriminate between DHPS and DOHH inhibitors.”

We would like to thank the reviewer for this very interesting point. To meet the reviewer request we have tested the Hpu98 antibody in the Hyp’Assay to determine if it could be used to detect only eIF5A deoxyhypusination. Experiments were performed as described for the Hyp’Assay but without DOHH in the reaction mix. First, we performed a dose response experiment with spermidine at pH 8.0 and 9.0 (new Fig. 3E). The Hyp’Assay allows the detection of deoxyhypusinated eIF5A. Under these conditions, hypusination was equally efficient at both pH tested. This suggests that the decrease in activity at pH 9.0, observed in Fig. 2D, is probably due to a decrease in activity of DOHH at pH 9.0. We also tested the ability of GC7 (new Fig. 3F) and CPX or EDTA (new Fig. 3G) to inhibit DHPS. As expected, only GC7 was efficient in inhibiting deoxyhypusination. As indicated by the reviewer this is another way to discriminate between DHPS and DOHH inhibitors. This is now described in lines 246-263.

2 .As mentioned above there are Hpu24 and Hpu98 antibodies available. The citation of the study describing them must be included 10.1016/j.jmb.2016.01.006

We agree with the reviewer, this is now reference 18 in our manuscript and is cited in line 127, 249 and 424.

3. The optimal pH for the assay was tested and found to be 8.0. The optimal pH for DHPS-catalyzed reaction seems to be above 9.2 (Park et al). In the case of using Hpu98 and using Hyp’assay to assess the DHPS activity it is necessary to check pH dependency

 This has been answered in our answer to comment 1.

4. Authors noticed that putrescine but not spermine has an inhibitory effect. Spermine is the substrate for the oxidoreductase reaction catalyzed by DHPS. As such, it may be possible that spermine is a donor of aminobutyl moiety to deoxyhypusinate eIF5a. Authors should challenge this using spermine as a substrate in Hyp’assay.

 This is a very interesting point. To answer this, we performed a dose response experiment of hypusination with spermidine or spermine. We observed that spermine can be used as a substrate by DHPS, although with less efficiency than spermidine. This is consistent to what has been published by Wator et al. (ref 23 in our manuscript). Under physiological conditions however, with spermine concentration in the micromolar range and the difference of affinity between spermine and spermidine, the contribution of spermine to hypusination should be low. These results have been added in the revised version of our manuscript in Supplementary Fig. 3 and are commented lines 175-182.

5. 5. Recently another study employed hypusine-specific antibodies to dectect hypusination (10.1111/febs.17207) in parasites. Authors should relate to this study. Also, this raises the possibility of using Hyp’assay to detect inhibitors of the hypusination in parasites. There are several attempts to develop such inhibitors in B. malay, L. donovani or other organisms such as tick. The authors should at least discuss such a possibility for their assay.

 This is again a very interesting point that we forgot to mention in our manuscript. The use of Hpu24 and Hpu98 to discriminate between deoxyhypusinated and hypusinated eIF5A by Western blot has been added in line 249-252 and the reference has been added (ref 30). A paragraph has been added lines 420-429 concerning the use of the Hyp’Assay to find inhibitors against DHPS/DOHH from parasite. 

6. In the introduction some important citations are missing 10.1016/j.str.2024.03.008 and 10.1038/s41467-023-37305-2 – studies describing the structural basis of deoxyhypusination in archaea and humans.

 We agree with the reviewer, that was an unfortunate oversight in our manuscript. A paragraph on eIF5A – DHPS complex has been added in our manuscript (lines 40-49, references 7 and 8).

Reviewer #3

The manuscript Development of a reliable, sensitive, and convenient assay for the discovery of new eIF5A hypusination inhibitors is an excellent attempt by the Peraldi et al and the results obtained are promising and can be very much beneficial for the readers of the journal, the authors have four new GC7 analogs and characterized them through various techniques and prove the structural assignments through these studies. The overall quality of the manuscript is nice and can be a good addition in the recent available literature of the field. However there are some old/irrelevant citations which should be replace, after careful reading I suggest the following replacements

replace reference number 5, 6 and 7 with

doi: https://doi.org/10.1016/j.ejso.2020.02.034

https://doi.org/10.1007/s00604-023-05652-y

https://doi.org/10.1021/acsaelm.4c00941

With this minor change I recommend this manuscript for publication

We would like to thank the reviewer for her/his kind appreciation of our work. 

As previously indicated to PLOS one (Case Number: 08657412) we would gladly make the proposed changes, however, it appears that some mistake has been performed in the copy and paste of the papers suggested by the reviewer.

This is a copy of the message we sent to PLOS one office:

We would gladly make the suggested modification, however it appears that the references provided by the reviewer are not related to our manuscript.

Our references 5-7 concern the biological function of the translation factor eIF5a.

The title of the first paper ( https://doi.org/10.1016/j.ejso.2020.02.034 ) is

“The impact of circumferential tumour location on the clinical outcome of rectal cancer patients managed with neoadjuvant chemoradiotherapy followed by total mesorectal excision”

This manuscript is about surgery of colorectal cancer.

The title of the second paper (https://doi.org/10.1007/s00604-023-05652-y) is

“Bi-enzyme competition based on ZIF-67 co-immobilization for real-time monitoring of exocellular ATP.

This manuscript is about the measure of extracellular ATP.

The title of the third paper ( https://doi.org/10.1021/acsaelm.4c00941 ) is

Ultrasensitive Photoelectric Immunoassay Platform Utilizing Biofunctional 2D Vertical SnS2/Ag2S Heterojunction.

This manuscript is about photoelectric immunoassay.

We feel that a wrong copy and paste has been performed.

Would it be possible to contact reviewer #3 to have the correct references so that we can add them in our revised manuscript?

Reviewer #4: 

The authors presented a convenient cell-free assay, Hyp’Assay, to monitor the hypusination of the translation factor eIF5A, and they performed screening of four GC7 analogs as new hypusination inhibitors they synthesized. It is interesting. However, I am confused with a major issue. It is suggested that the blockade of eIF5A hypusination limited cancerous cell growth in colorectal cancer, which needs further validation in the breast cancer. Why did the authors select a breast cancer cell line MCF-7, but not a classical colorectal cancer cell line, to test the reliability of the Hyp’Assay? The test with human colon cancer cell lines should be performed.

We would like to thank the reviewer for her/his kind assessment of our work. To meet her/his comment we have studied the effect of our compounds on the hypusination of eIF5A

---

## [Decision Letter · Decision Letter 1]

22 Oct 2024

Development of a reliable, sensitive, and convenient assay

 for the discovery of new eIF5A hypusination inhibitors

PONE-D-24-29255R1

Dear Dr. Peraldi,

We’re pleased to inform you that your manuscript has been judged scientifically suitable for publication and will be formally accepted for publication once it meets all outstanding technical requirements.

Kind regards,

Jorddy Neves Cruz

Academic Editor

PLOS ONE

Additional Editor Comments (optional):

Reviewers' comments:

Reviewer's Responses to Questions

**Comments to the Author**

1. If the authors have adequately addressed your comments raised in a previous round of review and you feel that this manuscript is now acceptable for publication, you may indicate that here to bypass the “Comments to the Author” section, enter your conflict of interest statement in the “Confidential to Editor” section, and submit your "Accept" recommendation.

Reviewer #1: All comments have been addressed

Reviewer #2: All comments have been addressed

Reviewer #4: All comments have been addressed

2. Is the manuscript technically sound, and do the data support the conclusions?

Reviewer #1: Yes

Reviewer #2: Yes

Reviewer #4: Yes

3. Has the statistical analysis been performed appropriately and rigorously? 

Reviewer #1: Yes

Reviewer #2: Yes

Reviewer #4: Yes

4. Have the authors made all data underlying the findings in their manuscript fully available?

Reviewer #1: Yes

Reviewer #2: Yes

Reviewer #4: Yes

5. Is the manuscript presented in an intelligible fashion and written in standard English?

Reviewer #1: Yes

Reviewer #2: Yes

Reviewer #4: Yes

6. Review Comments to the Author

Reviewer #1: The authors have adequately responded to all comments, and have modified the text in line with the suggestions.

Reviewer #2: The authors have addressed all my concerns, and the additional experiments significantly strengthen the manuscript's conclusions. I am satisfied with the revised version and fully support its publication in PLOS One. Additionally, regarding Reviewer 1's comments, while I understand the concerns raised, the authors should emphasize the superiority of their method. However, I agree with the authors that the title should remain "Development of..." rather than "Adaptation of...". The article by Wator et al uses antihypusine antibodies in a Western blot approach, not in an activity assay per se. As I mentioned in my review, this newly developed assay is highly anticipated, especially after the revisions, which simplify it for screening DHPS inhibitors by omitting DOHH from the assay. I look forward to seeing this screening tool tested by the research community.

Reviewer #4: The effect of the compounds on the hypusination of eIF5A in HT29 cells has been shown in new Fig. 5C. I recommend publication.

7. PLOS authors have the option to publish the peer review history of their article (what does this mean?). If published, this will include your full peer review and any attached files.

Reviewer #1: **Yes: **Per Sunnerhagen

Reviewer #2: No

Reviewer #4: No

---

## [Editor Report · Acceptance letter]

30 Oct 2024

PONE-D-24-29255R1 

PLOS ONE

Dear Dr. Peraldi, 

I'm pleased to inform you that your manuscript has been deemed suitable for publication in PLOS ONE. Congratulations! Your manuscript is now being handed over to our production team.

Kind regards, 

on behalf of

Dr. Jorddy Neves Cruz 

Academic Editor

PLOS ONE